Resource

# Systematic assessment of structural variant annotation tools for genomic interpretation

Xuanshi Liu[1], Lei Gu[2], Chanjuan Hao[1], Wenjian Xu[1], Fei Leng[1], Peng Zhang[1], Wei Li[1]

**Structural variants (SVs) over 50 base pairs play a significant role in phenotypic diversity and are associated with various diseases, but their analysis is complex and resource-intensive. Numerous computational tools have been developed for SV prioritization, yet their effectiveness in biomedicine remains unclear. Here we benchmarked eight widely used SV prioritization tools, categorized into knowledge-driven (AnnotSV, ClassifyCNV) and data-driven (CADD-SV, dbCNV, StrVCTVRE, SVScore, TADA, XCNV) groups in accordance with the ACMG guidelines. We assessed their accuracy, robustness, and usability across diverse genomic contexts, biological mechanisms and computational efficiency using seven carefully curated independent datasets. Our results revealed that both groups of methods exhibit comparable effectiveness in predicting SV pathogenicity, although performance varies among tools, emphasizing the importance of selecting the appropriate tool based on specific research purposes. Furthermore, we pinpointed the potential improvement of expanding these tools for future applications. Our benchmarking framework provides a crucial evaluation method for SV analysis tools, offering practical guidance for biomedical research and facilitating the advancement of better genomic research tools.**

## Introduction

Structural variants (SVs), namely genetic alterations exceeding 50 base pairs (bp), significantly contribute to phenotypic diversity and underlie the mechanisms of a wide spectrum of human disorders, from rare diseases such as thrombocytopenia-absent radius syndrome (Klopocki et al, 2007) to common ones like autism spectrum disorder (Zhang et al, 2023) and cancer (Li et al, 2020). However, SVs represent a diverse spectrum of genomic changes containing deletions, duplications, inversions, insertions, translocations, and more complex variations (Collins et al, 2020), which present significant challenges for detection and analysis.

Detecting SVs using short-read sequencing poses challenges due to difficulties in aligning reads and accurately determining the full genomic span affected by an SV, especially when breakpoints occur within tandem repeats or involve sequences absent from the reference genome. Although long-read sequencing can mitigate some of these challenges by providing longer and more contiguous reads, it is often constrained by higher costs, lower throughput, and increased error rates compared with short-read sequencing. In addition, the vast number of SVs detected, thousands through short-read and up to 20,000 through long-read whole genome sequencing (WGS) (Collins et al, 2020; Beyter et al, 2021), results in the complexity of their analysis and interpretation.

The functional impact of SVs is complex, directly influencing gene function and indirectly affecting regulatory regions through long-range interactions (Lupianez et al, 2015). Moreover, a significant portion of SVs is found in noncoding regions, where our understanding is still evolving. Traditional methods for assessing the functionality or causality of SVs, such as association studies and eQTL analysis, require extensive cohorts, high-throughput sequencing, and sophisticated data analysis (Brandler et al, 2018). Family based studies, while valuable, are resource-intensive with specialized expertise (Pagnamenta et al, 2023).

Given the complexity and the high number of SVs, computational tools for their prioritization have become essential. Since 2015, more than two dozen tools have been introduced, predominantly in the last 3 yr. However, there is currently no study in comparing the performance of these SV prioritization tools. To fill this gap, we have selected eight tools for benchmarking based on their availability, periodic updates, ability to handle various SV types without additional information or manual work, and computational efficiency in terms of computational resource usage and compatibility with standard pipelines (Table S1).

These eight tools are categorized into two types: the first type, or knowledge-driven, such as AnnotSV (Geoffroy et al, 2021) and ClassifyCNV (Gurbich & Ilinsky, 2020), is based on established clinical evaluation guidelines from the American College of Medical Genetics and Genomics (ACMG) and the Clinical Genome Resource

[1]Beijing Key Laboratory for Genetics of Birth Defects, Beijing Pediatric Research Institute; MOE Key Laboratory of Major Diseases in Children; Genetics and Birth Defects Control Center, National Center for Children's Health; Beijing Children's Hospital, Capital Medical University, Beijing, China  [2]Epigenetics Laboratory, Max-Planck Institute for Heart and Lung Research, Cardiopulmonary Institute, Bad Nauheim, Germany

Correspondence: liwei@bch.com.cn

**Table 1.   Overview of the approaches evaluated in this work.**

| Software[a] | Year | Main language | Assumption | Classifier | Training set | Result | URL |
|---|---|---|---|---|---|---|---|
| AnnotSV (Version: 3.3.6) | 2018 | Tcl, Shell, Python | ACMG | ACMG | Implementation of ACMG guideline. | Annotation, scores | https://lbgi.fr/AnnotSV/ |
| CADD-SV (Version 1.1) | 2022 | Python, R | Evolutionary fitness | Random forest | Randomly distributed SVs over the human autosomes, evolutionarily fixed chimpanzee and human-derived SVs. | Scores | https://cadd-sv.bihealth.org/ |
| ClassifyCNV (Version 1.1.1) | 2020 | Python, Shell | ACMG | ACMG | Implementation of ACMG guideline. | Scores | https://github.com/Genotek/ClassifyCNV |
| dbCNV | 2023 | Perl, Shell | Molecular functions | Gradient boosted trees | The ClinVar, dbVar, ClinGen, DGV, DECIPHER and gnomAD (accessed before January 2023) | Classification | https://github.com/lllllv-1/dbCNV |
| StrVCTVRE (Version 1.7) | 2022 | Python | Molecular functions on exons | Random forest | Rare SVs from ClinVar, gnomAD, and a recent great ape sequencing study. | Scores | https://strvctvre.berkeley.edu/ |
| SVScore (Version 0.6) | 2017 | Perl, Shell | SNPs-based CADD scores | Derived from CADD[b] | The precomputed SNP scores generated by CADD v1.3 | Scores | https://github.com/lganel/SVScore |
| TADA (Version 1.0.2) | 2022 | Python, Shell | Molecular functions related to long range interaction | Random forest | DECIPHER, Variants in the set published by Audano et al (2019), GnomAD, UK Biobank data set and DGV. | Scores | https://github.com/jakob-he/TADA/ |
| XCNV | 2022 | R, Shell | Molecular functions | XGBoost | The dbVar, ClinGen, DECIPHER v10.1, and DGV (accessed before January 2021). | Scores | https://github.com/kbvstmd/XCNV |

[a]Software version was given if available.
[b]CADD was generated by the support vector machine.

(ClinGen), which serve as the gold standard for the clinical evaluation and etiological diagnosis of genetic disorders (Richards et al, 2015). The second type, or data-driven, including tools such as CADD-SV (Kleinert & Kircher, 2022), dbCNV (Lv et al, 2023), StrVCTVRE (Sharo et al, 2022), SVScore (Ganel et al, 2017), TADA (Hertzberg et al, 2022), and XCNV (Zhang et al, 2021), employs machine learning models such as random forest, gradient boosted trees, and XGBoost to estimate SV effects, differing in features or training sets.

The knowledge-driven approaches implemented related databases described in ACMG guidelines stratified by SV types, incorporating factors like protein-coding or other functionally important elements, gene numbers, haploinsufficiency, benign regions, and inheritance patterns. In contrast, data-driven approaches based their training sets and features on gold standard datasets, including ClinVar (Landrum et al, 2016), DECIPHER (Firth et al, 2009), DGV (MacDonald et al, 2014), GnomAD (Collins et al, 2020), and 1 KG (1000 Genomes Project Consortium et al, 2015), with a focus on specific aspects of SV analysis. For example, CADD-SV used training sets derived from human and chimpanzee SVs as neutral proxies, whereas dbCNV incorporated diverse gold standard datasets within its scoring models. StrVCTVRE focused on molecular functions overlapping exons, SVScore aggregated scores from individual SNPs, TADA considered long-range hypotheses from 3D genomic

data, and XCNV integrated a broad spectrum of population genomic information.

In this study, we evaluated the eight SV prioritization approaches in accuracy, robustness, and usability of SV across various genomic contexts and biological backgrounds. We hope to provide a comprehensive evaluation to assist researchers and clinicians in choosing the most appropriate tools for their study purposes or dataset usage. Furthermore, we discuss the future directions of SV prioritization approaches, offering insights into the field to facilitate the development of tools.

# Results

## Description of benchmarking pipeline

In our systematic evaluation (Table S1), we identified eight computational approaches developed between 2017 and 2023: AnnotSV, CADD-SV, ClassifyCNV, dbCNV, StrVCTVRE, SVScore, TADA, and XCNV (Table 1). The knowledge-driven approaches, AnnotSV and ClassifyCNV which included scoring metrics, demanded considerable expertise for implementation based on ACMG criteria. In contrast, data-driven approaches primarily generated scores to prioritize SVs.

**Table 2.  Summary of seven independent datasets used in this study.**

| Benchmark dataset | Positive set (number of positive variants) | Negative set (number of negative variants) |
|---|---|---|
| Germline SVs[a] from ClinVar and GnomAD | "pathogenic" and "likely pathogenic" germline SVs from ClinVar (January. 2023–April. 2024) (N = 489). | (1) "benign" and "likely benign" germline SVs from ClinVar (January. 2023–April. 2024) (N = 93); (2) randomly select rare SVs with matched lengths with positive sets from gnomAD v4 (N = 396). |
| Noncoding SVs and GnomAD | Noncoding SVs from peer-reviewed publications (N = 6). | Randomly select rare SVs with matched lengths with positive sets from gnomAD v4; No overlapped with protein coding genes listed at gencode v30lift37 (N = 6). |
| Long range SVs and GnomAD | SVs implicated in long-range interactions from peer-reviewed publications (N = 12). | Randomly select rare SVs with matched lengths with positive sets from gnomAD v4 (N = 12). |
| Somatic SVs | Somatic SVs from COSMIC (v99) with recurrence >=2 and located on risk genes listed at oncoKB (N = 218). | Randomly select somatic SVs from COSMIC (v99) with recurrence = 1 and no overlapped with risk genes listed at oncoKB (N = 238). |
| Disease associated SVs from a GWAS and GnomAD | Rare SVs which validated by replication listed at the peer-reviewed publication (N = 32). | Randomly select rare SVs with matched lengths with positive sets from gnomAD v4 (N = 32). |
| Functional relevant SVs from eQTL studies and GnomAD | Rare SVs: aberrant gene expression is in multi tissues and the gene has dosage changed (N = 72). | Randomly select rare SVs with matched lengths with positive sets from gnomAD v4 (N = 72). |

[a]SVs including CNVs and deletions, duplications.

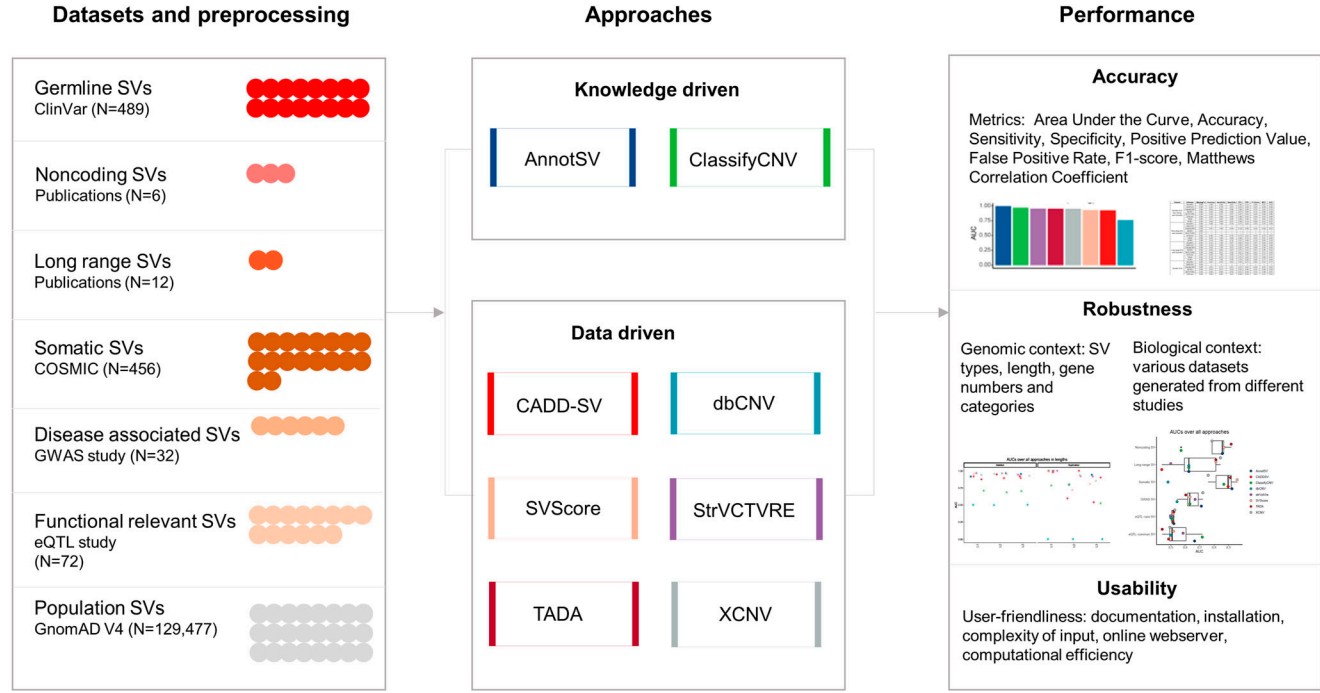

**Figure 1.  Overview of study workflow for SV prioritization benchmarking.**
This workflow illustrates the evaluation process for eight SV prioritization tools, categorized into knowledge-driven and data-driven approaches. These tools were benchmarked across seven independent and curated datasets using three main criteria: (1) accuracy in pathogenicity prediction, (2) robustness in diverse genomic and biological contexts, and (3) usability, focusing on user accessibility and computational performance.

Our benchmarking used six datasets constructed from seven different data sources, with GnomAD serving as a negative control set (Tables 2 and S2). The datasets encompassed a total of 489 germline SVs from ClinVar, six noncoding SVs, 12 long-range SVs, 456 somatic SVs from COSMIC (Sondka et al, 2024), 32 GWAS SVs, and 72 eQTL SVs. The performance of these approaches was assessed based on three key criteria: accuracy, robustness, and usability (Fig 1). Accuracy was evaluated using the AUC metric on the ClinVar dataset since the ability to identify pathogenic SVs. Robustness was examined in the context of genomic and biological variability. Usability was measured by computational efficiency and the user-friendliness of the tools, including the

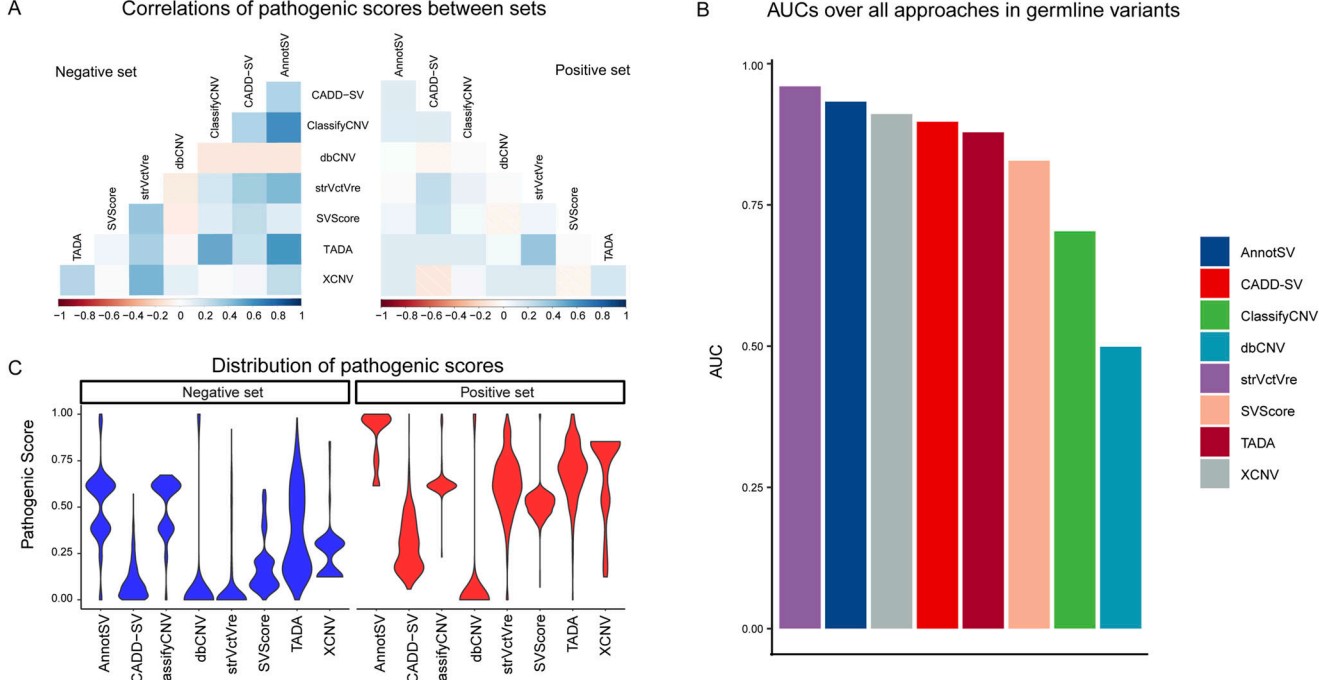

**Figure 2. Comparative performance of eight SV prioritization approaches.**
**(A)** Correlation analysis between positive (pathogenic) and negative (benign) variant sets across the eight approaches, indicating the differentiation ability of each tool. **(B)** Distribution of pathogenicity scores for positive and negative sets, showing score separation across the tools. **(C)** Performance summary across all germline variants from ClinVar, measured by area under the curve.

quality of documentation, ease of installation, requirements of preinstalled datasets, complexity of input files, and the presence of an online webserver.

### Benchmarking performance evaluation of accuracy

Our comprehensive evaluation revealed significant variability in the predictive concordance among the eight SV prioritization approaches. Spearman rank correlation coefficients indicated a higher degree of consistency for the negative set compared with the positive set, with weak correlations (R < 0.3) prevalent among the approaches (Fig 2A). This observation suggests a lack of consensus in predictive capabilities, underscoring the necessity for a thorough comparative assessment.

In assessing accuracy using the AUC metric against gold standard datasets, StrVCTVRE stood out with an AUC of 0.96, demonstrating exceptional performance (Fig 2B, Table S3). Within the data-driven approaches, XCNV, CADD-SV, TADA, and SVScore also exhibited commendable AUCs ranging from 0.91 to 0.83. Conversely, dbCNV showed a notably lower performance with an AUC of 0.50. The distribution of pathogenic score (PS) for positive and negative sets was distinctly separable in most data-driven methods, whereas dbCNV showed overlapping distributions (Fig 2C). Knowledge-driven models, AnnotSV and ClassifyCNV, also performed relatively well, with AUCs of 0.93 and 0.70, respectively. These results highlight the competitive performance

of both knowledge-driven and data-driven models, particularly StrVCTVRE and AnnotSV.

### Performance evaluation of the robustness on genomic features

We conducted a robustness evaluation of the approaches based on genomic features, including SV types, lengths, and gene contents. According to ACMG guidelines, deletions and duplications were assessed separately. The performance of most approaches was found to be similar for both SV types, which aligned with the distribution of PS (Fig 3A). StrVCTVRE, AnnotSV, XCNV, CADD-SV, and ClassifyCNV demonstrated AUCs above 0.71 for deletions and 0.63 for duplications (Fig 3B, Table S4). However, TADA, SVScore, and dbCNV were less consistent, especially for duplications, where their AUCs were considerably lower.

When assessing SV performance across different length ranges $(>6*10^3, 6*10^3{\sim}10^5, > 10^5)$ (Fig 3C), StrVCTVRE, AnnotSV, XCNV, CADD-SV, and TADA maintained high and consistent performance (AUCs > 0.80) in deletions across all size groups (Fig 3D, Table S5). In contrast, ClassifyCNV and dbCNV showed relatively poor performances, and SVScore displayed a lower AUC (AUC = 0.65) for lengths greater than $10^5$ bp. For duplications, a decline in performance with increasing length was observed, particularly for TADA, which showed a decline in AUC from 0.98 for shorter duplications to 0.94 for longer ones. This trend may be attributed to the size-match strategy used in TADA's training set construction. ClassifyCNV and SVScore showed less promising performance for longer

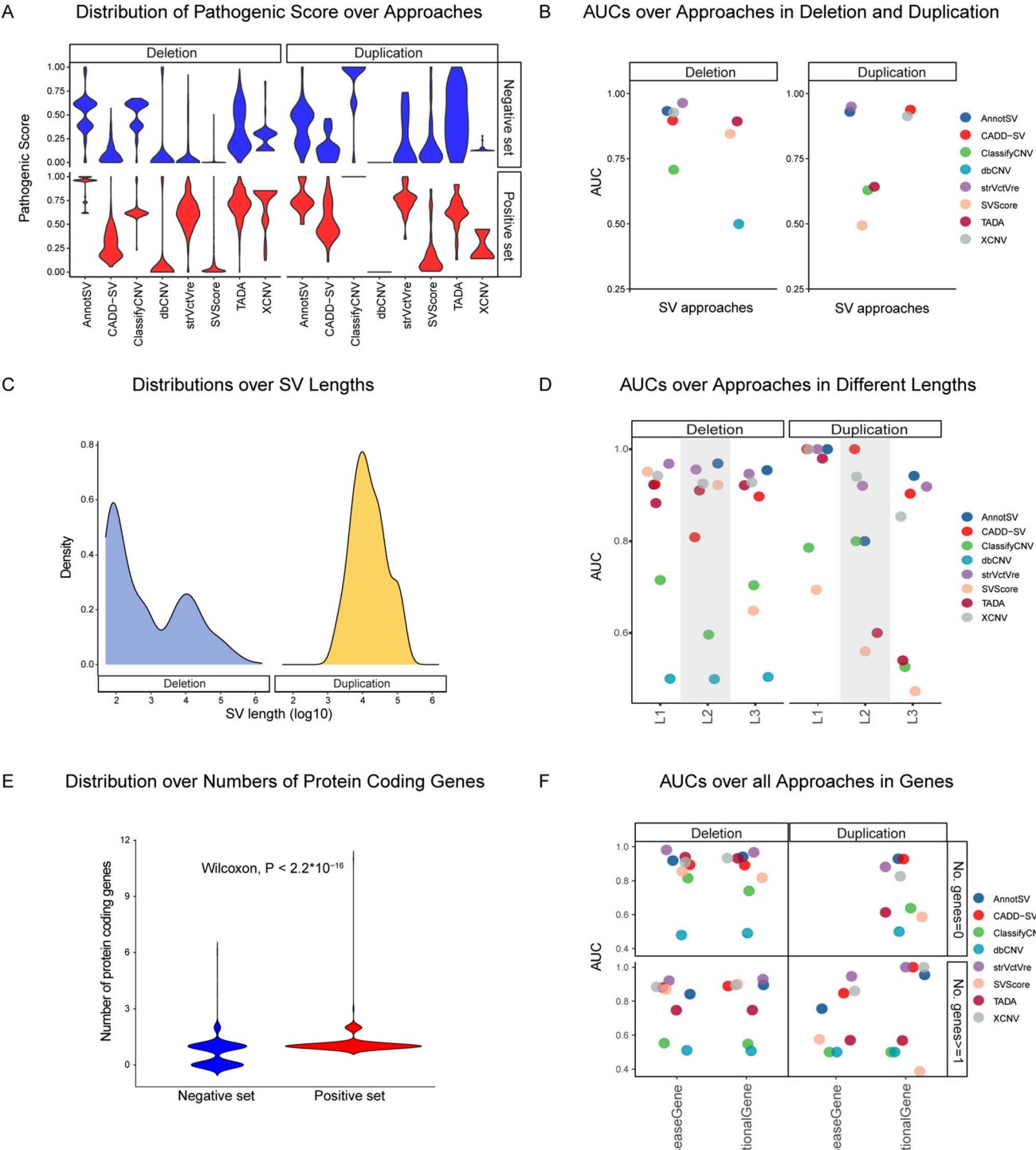

**Figure 3. Performance of SV prioritization tools across genomic contexts.**
SV type, length, and gene content: **(A)** Distribution of pathogenicity scores for deletion and duplication sets, illustrating score separation across tools by SV type. **(B)** Performance of each tool in deletions and duplications among germline ClinVar variants, evaluated by area under the curve (AUC). **(C)** Length distributions of deletions and duplications within the dataset. **(D)** AUCs performance over three lengths ranges (L1< $6*10^3$, L2:$6*10^3$~$10^5$, L3 >$10^5$) for deletions and duplications. **(E)** Distribution differences in protein-coding gene coverage between negative (benign) and positive (pathogenic) SV sets. **(F)** AUC comparison by gene context (disease-related, functional genes) for deletions and duplications, further categorized by SVs covering zero genes (No. genes = 0) and one or more genes (No. genes ≥ 1). AUC, area under the curve; SV, structural variant.

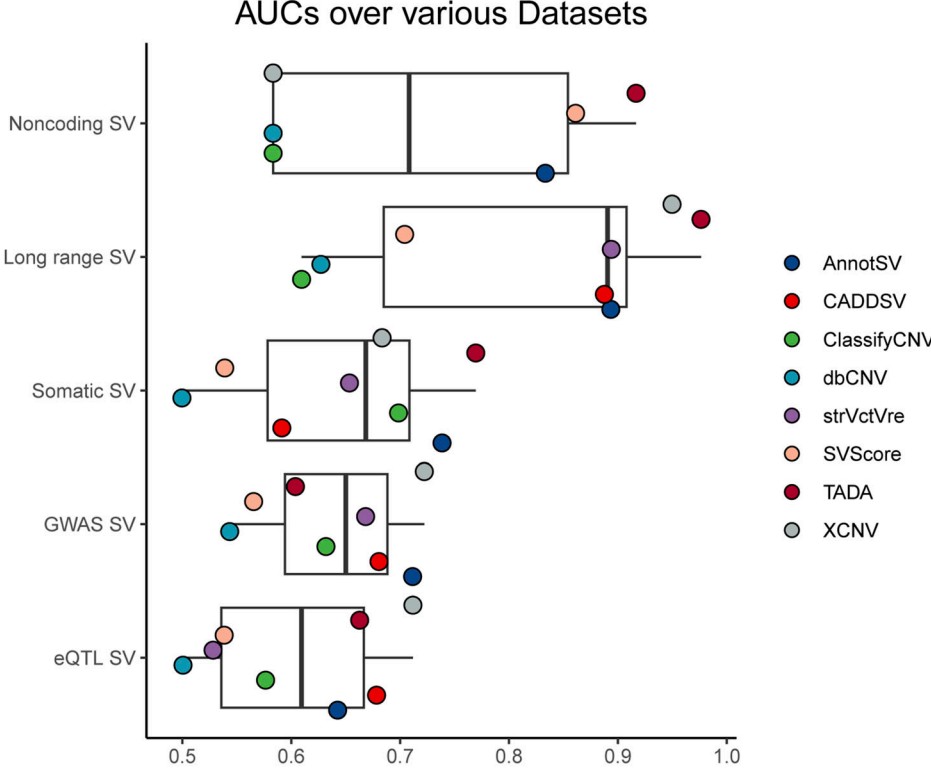

**Figure 4. Performance across approaches covering various biological mechanisms including noncoding SVs, long range SVs, somatic SVs, GWAS SV and eQTL SV.**
AUC, area under the curve; SV, structural variant.

duplications, where dbCNV failed to distinguish between positive and negative sets. These results demonstrated that CADD-SV, AnnotSV, StrVCTVRE, and XCNV had high efficacy across various SV length groups for both deletions and duplications, while TADA, SVScore, and ClassifyCNV exhibited diverse performances, particularly for longer lengths (Fig 3D, Table S5).

From the comparison of SV types and length groups, we observed that the distribution of PS in the duplication set was generally less distinct. This may be due to the smaller number of duplications in the training set and feature selection processes that were more tailored to deletions. The predominance of shorter deletions over duplications in pathogenic status requires careful consideration. For example, the commonly used training set ClinVar includes 11,946 germline pathogenic deletions with an average length of 122,698 bp and 1,391 duplications with an average length of 131,202 bp, with over 13% and 15% of deletions and duplications being longer than the average length, respectively (Fig 3C).

Gene content analysis revealed significant differences between the number of protein-coding genes covered by SVs in negative and positive sets (Figs 3E and S1). CADD-SV, AnnotSV, StrVCTVRE, and XCNV consistently showed superior performance across different gene content categories, irrespective of the number of genes involved, with AUCs exceeding 0.85 (Fig 3F, Table S6). In contrast, TADA, SVScore, and ClassifyCNV performed better in deletions not associated with any disease or functional genes. Notably, deletions without disease or functional genes were longer but not significantly (mean length: 20,948.65 bp). For duplications, the performance of CADD-SV,

AnnotSV, StrVCTVRE, and XCNV remained high in groups intersecting with at least one disease or functional gene (Fig 3F, Table S6). In summary, our study indicates that while most approaches exhibit improved performance in the absence of disease or functional genes in deletions, their efficacy varies in duplications.

Collectively, StrVCTVRE, AnnotSV, CADD-SV, and XCNV have demonstrated superior performance across various metrics, including SV types, length groups, and gene contents, indicating their robustness in predicting SV pathogenicity.

## Performance evaluation of the robustness on biological mechanisms

Our investigation into the robustness of computational approaches for SV analysis extended to examining biological mechanisms. We systematically curated five distinct datasets representing a spectrum of genomic variations, including noncoding SVs, long-range SVs, somatic SVs, disease-associated SVs, and functionally relevant SVs.

In noncoding SVs, we observed that TADA, SVScore, and AnnotSV were the top performers, demonstrating high AUC values of 0.92, 0.86, and 0.83, respectively (Fig 4). These tools showed strong alignment between AUC and other performance metrics such as accuracy, sensitivity, and specificity (Table S3). However, it is important to note that CADD-SV and StrVCTVRE were not applicable for noncoding SVs due to their focus on protein-coding genes. Long range SVs were evaluated, revealing TADA and XCNV as standout performers with AUCs of 0.98 and 0.95, respectively (Fig 4).

**Table 3. Summary of computational efficiency and user-friendliness over all approaches.**

| Software | Knowledge driven | | Data driven | | | | | |
|---|---|---|---|---|---|---|---|---|
| | AnnotSV | ClassifyCNV | CADD-SV | dbCNV | StrVCTVRE | SVScore | TADA | XCNV |
| Efficiency (second) | 12 | 3 | 100 (20 cores) | 38 | 19 | 5 | 16 | 24 |
| Document quality | Excellent | Normal | Good | Normal | Good | Normal | Good | Good |
| Installation | cmd, conda, docker | cmd | conda | cmd | cmd, conda | cmd | cmd | cmd |
| Prerequisite dataset | Yes | Yes | Yes | No | Yes | Yes | Yes | Yes |
| Genome build | hg19, hg38 | hg19, hg38 | hg38 | hg19 | hg19, hg38 | hg19 | hg19 | hg19 |
| Input | Bed, vcf | Bed | Bed | Bed | Bed | vcf | Bed | Bed |
| SV types | Deletion, insertion, duplication, inversion, breakend record | CNV | Deletion, insertion and duplication | CNV | Deletion, duplication | All types | CNV | CNV |
| Online webserver | Yes | No | Yes | No | Yes | No | No | Yes |

StrVCTVRE, AnnotSV, and CADD-SV also exhibited robust performance, albeit with StrVCTVRE's results being partially obscured by an 11% variant missing rate (Table S3). The discordance between SVScore's high AUC and its lower MCC suggested a high false positive rate, likely due to its default scoring system for long SVs.

Somatic SVs were assessed with TADA and AnnotSV leading the way with AUCs of 0.77 and 0.74, respectively (Fig 4). Regarding to MCC, the knowledge-driven approaches, particularly AnnotSV and ClassifyCNV, showed a slight advantage over data-driven methods. When assessing disease-associated SVs from large cohort studies, all methods except SVScore showed comparable AUCs and MCCs (Fig 4, Table S3). However, CADD-SV, StrVCTVRE, and TADA exhibited missing variant rates, indicating a need for improvement in detecting these variants.

Finally, the analysis of functionally relevant SVs revealed varying performance among data-driven approaches. XCNV emerged as the top performer with an AUC of 0.71, followed by CADD-SV and TADA (Fig 4). In contrast, dbCNV and StrVCTVRE lagged behind, highlighting challenges in accurately predicting these SVs.

In summary, our assessment confirmed the potential of these tools to identify novel biological mechanisms from germline variants. While the tools may not exhibit the highest level of AUC for somatic, GWAS, and eQTL SVs, their performances provide a foundation for further refinement. Our findings showed the importance of selecting the appropriate tool based on the specific characteristics of the SVs and highlight the potential for further refinement across various genomic contexts.

### Usability evaluation of computational efficiency and user-friendliness

The usability of the approaches was assessed. We focused on computational efficiency and user-friendliness, which significantly impact user experience and practical applicability (Table 3). Computational efficiency revealed that knowledge-driven approaches generally outperformed data-driven approaches with the completing tasks within an average of 15 s. ClassifyCNV was notably efficient, while StrVCTVRE and TADA led among data-driven methods. Notably, the use of default hyperparameter settings during testing influenced method efficiency. For instance, CADD-SV provides multicore operational capability, which may influent efficiency.

Regarding the quality of tutorials and code, we found that most methods adequately met the basic requirements of users, ensuring straightforward installation and use. Several approaches offered support through conda environment and Docker images, which greatly facilitated the setup process. However, the necessity to install datasets is a speed limit step which was dependent on internet connection stability. Our analysis also considered the complexity of input files, including supported genome builds, file types, and SV types. We noted that the most of tools supported at least two SV types. The hg19 genome build was commonly accepted, though hg38 is increasingly adopted. The bed format, specifying chromosome, start position, end position, and SV type, emerged as standard among the evaluated tools. In addition, four out of the eight approaches provided an online version, including AnnotSV, CADD-SV, StrVCTVRE, XCNV, enhancing accessibility and user convenience.

## Discussion

In the landscape of genomic sequencing, computational methods have become indispensable for deciphering the functional relevance and clinical significance of SVs. We created seven datasets from diverse biological backgrounds and conducted an extensive benchmarking of eight available approaches, categorized into knowledge-driven and data-driven, focusing on accuracy, robustness, and usability. Our findings reveal that both categories of tools demonstrate comparable effectiveness in identifying pathogenic germline SVs. This study systematically evaluates and compares the

performance of SV prioritization tools, offering important insights to the biomedical and clinical communities.

Our evaluations yielded several key insights. First, the comparable effectiveness in identifying pathogenic germline SVs across different methods suggests that the choice between these approaches should be guided by the specific context and objectives of the analysis rather than any inherent superiority. This is a significant hint that underscores the need for a contextual approach in selecting SV analysis tools. In addition, our benchmarking study highlights the strengths and limitations of both knowledge-driven and data-driven techniques. Future tools could benefit from a hybrid approach. Knowledge-based techniques which leverage existing knowledge and framework like the ACMG guidelines, are essential for determine the pathogenicity of SVs. Incorporating data-driven techniques can be highly beneficial in identifying novel or potentially pathogenic SVs that may not be well understood yet. Integrating both approaches can lead to more comprehensive and accurate SV prioritization, especially for novel or complex regions.

Second, the capacity of these methods to integrate new knowledge and generate new hypotheses is critical. For example, the importance of small variants in noncoding regions is well-established, as illustrated by examples such as a variant in the promoter region of *GATA1* affecting a transcription factor binding site, leading to hereditary persistence of fetal hemoglobin (Martyn et al, 2019), or a variant disrupting upstream open reading frames of the *NF2* gene causing neurofibromatosis type 2 (Whiffin et al, 2020). With WGS uncovering hundreds of thousands of SVs, primarily impacting noncoding regions, the ability of these tools to accommodate emerging data is essential for scientific discovery.

Third, the applicability of these methods to variants beyond germline SVs is highly significant. The performances are acceptable for initial screening and can be particularly useful in data generation or in settings where a broader filter is applied to capture potential variants of interest. Recently, several studies focused on discovery the somatic variants from whole exome sequencing data from UK Biobank (Bernstein et al, 2024). As the understanding of the role of somatic and other non-germline variants in disease grows, tools capable of analyzing a broader spectrum of variants become increasingly important.

Despite these advancements, challenges persist in generating unified SV sets across all types, especially from short-read WGS. Most existing approaches concentrate on deletions and duplications, often overlooking other SV types. This limitation may stem from the developing status of ACMG guidelines and the scarcity of gold standard datasets for certain SV types. The increasing accessibility of long-read sequencing opens up new opportunities for SV detection. This technique is particularly effective for identifying complex SVs, repetitive regions, and resolving large structural changes that short-read technologies failed. However, it also faces challenges. These new regions will require updated annotations and retraining of data-driven models to handle the unique properties of long-read data. Moreover, integrating long-read sequencing data with the existing short-read data and annotations poses another challenge. There is a need for tools that can efficiently combine information from multiple sequencing platforms and provide a unified annotation framework.

Understanding the underlying biological mechanisms also necessitates integrating cell-type specific information and phenotype data (Liu et al, 2023; Sanchez-Gaya & Rada-Iglesias, 2023). Promisingly, recent methodologies have begun to incorporate phenotype-specific characteristics (Althagafi et al, 2022; Xu et al, 2023), recognizing their significance in assessing SV pathogenicity. A particular challenge lies in interpreting the biological significance of SVs within noncoding regions, where their impact often depends on disruptions to regulatory elements such as enhancer–promoter interactions and topologically associating domain (TAD) boundaries. Tools that incorporate 3D genomic context could improve noncoding SV interpretation (Hertzberg et al, 2022; Poszewiecka et al, 2022).

Finally, CHM13/T2T represents a major improvement in genome completeness, especially in difficult regions like centromeres and telomeres. Combining it with updated annotations and resources could be a promising direction for tool development, benefiting future clinical and biological studies. As the identification of pathogenic SVs increases, comprehensive annotation of the noncoding genome, a deeper understanding of SVs in disease etiology, and advancements in bioinformatics technologies will undoubtedly spur the development of additional tools. Our future work will compare these emerging tools as they become available.

In conclusion, our study provides a critical evaluation of computational tools for prioritizing SVs, highlighting their accuracy, robustness, and usability. The findings emphasize the importance of selecting tools based on the specific analysis context and objectives. As genomics continues to evolve, the adaptability of these tools to new knowledge and data generation will be crucial for advancing our understanding of the genomic basis of disease.

# Materials and Methods

### Dataset curation

To ensure the strength and reproducibility of our benchmarking assessments, the creation of an independent dataset is required that does not overlap with any variants used in the training datasets of the software under evaluation. Our pipeline created seven distinct datasets, with the first six serving as positive datasets to evaluate specific aspects of software performance, and the last one serving as a negative control which were: (1) germline SVs from ClinVar; (2) SVs in noncoding regions (Noncoding SVs); (3) SVs involved in long-range interactions (long range SVs); (4) somatic SVs from COSMIC (https://cancer.sanger.ac.uk/cosmic); (5) validated SVs from GWAS (Auwerx et al, 2024); (6) functionally relevant SVs from eQTL studies (Scott et al, 2021); (7) population SVs from GnomAD version 4.1 (https://gnomad.broadinstitute.org/) (Fig 1, Tables 2 and S2). Employing a dual-strategy approach, we ensured that the access date for our test datasets was subsequent to the publication date of the evaluated software and meticulously eliminated any overlapping SVs among the datasets.

The first benchmark dataset, derived from ClinVar in March 2024, focused on CNVs, including deletions and duplications, classified according to ACMG guidelines. We retained CNVs exceeding 50 bp in length and meeting specific classification criteria ("pathogenic," and "likely pathogenic" as positive labels, "likely benign" and "benign" as negative labels). To address the limited number of negative sets, we randomly selected deletions and duplications with allele frequencies less than 1% from GnomAD, confirming no overlap with GnomAD V2 or pathogenic SVs in ClinVar.

The second to fifth benchmark datasets were curated to reflect disease relevance with diverse biological origins (Table 2). Non-coding SVs with established pathogenicity were identified from peer-reviewed publications, emphasizing the role of noncoding regions in genetic pathology (Gordon et al, 2014; Bieth et al, 2015; Turner et al, 2016; Cappuccio et al, 2019) (Table S7). Long-range SVs were sourced from studies demonstrating their impact on the three-dimensional genome architecture (Kouwenhoven et al, 2010; Ellaway et al, 2013; Tayebi et al, 2014; Lupianez et al, 2015; Franke et al, 2016; D'Haene et al, 2019; Long et al, 2020) (Table S7). Somatic SVs were derived from COSMIC, and we constructed a matched positive and negative SV dataset following the approach by Wang et al (2023) and oncoKB (Chakravarty et al, 2017). Disease-associated SVs from GWAS were included based on validation and significance thresholds.

The sixth dataset from eQTL studies aimed to connect molecular and clinical phenotypes, focusing on rare SVs with aberrant gene expression across multiple tissues (Scott et al, 2021). The consequence of SV with respect to outlier gene is either complete dosage change or partial dosage change.

The final dataset, comprising population SVs, served as negative controls, with additional rare GnomAD variants added for comprehensiveness. All variants were lifted over to hg19 using UCSC liftover tool. We restricted our analysis to autosomes in hg19 genome build.

### Feature selection

Genomic content, crucial for evaluating the disease or functional relevance of SVs, was systematically compiled. We collected three groups of genes: protein-coding genes, disease-associated genes, and functionally relevant genes. Protein-coding genes were sourced from GENCODE. Disease associated genes were obtained from Orphanet (https://www.orpha.net/consor/cgi-bin/index.php), genes with dosage sensitivity from ClinGen (https://search.clinicalgenome.org/kb/gene-dosage/cnv) and ACMG-approved genes (V3.0). The functional relevant genes were collected among essential genes from cell culture studies (Hart et al, 2017), genes lethal in mouse models (Motenko et al, 2015), and genes with predicted dosage sensitive (probability of haploinsufficiency >= 0.9 or probability of triplosensitivity >= 0.9) (Collins et al, 2022). Annotations were based on distinct feature types in hg19 genome build (Liu et al, 2023).

### Workflow building and evaluation method

All methods, except dbCNV, generated pathogenic scores (PS) for SV prioritization with lower scores indicating non-pathogenicity and higher scores suggesting pathogenicity. For dbCNV, the five-tier classification was converted into numeric indicators. PS was derived using default parameters, followed by min–max normalization. All methods operated using default settings.

To evaluate the performance among approaches, we used a suite of metrics including accuracy (Equation (1)), sensitivity (Equation (2)), specificity (Equation (3)), positive prediction value (PPV) (Equation (4)), false positive rate (FPR) (Equation (5)), F1-score (Equation (6)), Matthews correlation coefficient (MCC) (Equation (7)) and area under the curve (AUC). Data visualization and analysis scripts were conducted using R and self-authored scripts.

$$Accuracy = (TP + TN)/(TP + FP + TN + FN) \tag{1}$$

$$Sensitivity = TP/(TP + FN) \tag{2}$$

$$Specificity = TN/(TN + FP) \tag{3}$$

$$PPV = TP/(TP + FP) \tag{4}$$

$$FPR = FP/(FP + TN) \tag{5}$$

$$F1 - Score = 2 * (PPV * Sensitivity)/(PPV + Sensitivity) \tag{6}$$

$$MCC = ((TP * TN - FP * FN)) \Big/ \sqrt{((TP + FP)(TP + FN)(TN + FP)(TN + FN))} \tag{7}$$

### Computational resource

The computational resources for testing all approaches including an Intel(R) Xeon(R) Gold 6140 CPU @ 2.30 GHz with 144 cores and 1 TB of memory, running CentOS Linux release 7.7.1908.

## Data Availability

The data accessed in this article are available in ClinVar (accessed at 2024-Mar-20), GnomAD (v4.1) and Cosmic (accessed at 2024-Mar-20).

## Supplementary Information

## Acknowledgements

This work was supported by the Ministry of Science and Technology of China [2019YFA0802104], the National Natural Science Foundation of China [32293204] to W Li; the Beijing Natural Science Foundation [5222007] to X Liu.

## Author Contributions

X Liu: conceptualization, data curation, formal analysis, funding acquisition, investigation, visualization, and writing—original draft, review, and editing.

L Gu: visualization, methodology, and writing—review and editing.

C Hao: supervision, project administration, and interpretation of the data.

W Xu: methodology and interpretation of the data.

F Leng: interpretation of the data.

P Zhang: validation, visualization, and interpretation of the data.

W Li: supervision, funding acquisition, project administration, and writing—review and editing.

## Conflict of Interest Statement

The authors declare that they have no conflict of interest.

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
