## [Reviewer comments · Life Science Alliance]

Life Science Alliance

Systematic Assessment of Structural Variant Annotation Tools for Genomic Interpretation

Xuanshi Liu, Lei Gu, Chanjuan Hao, Wenjian Xu, Fei Leng, Peng Zhang, and Wei Li

DOI: [10.26508/lsa.202402949](https://doi.org/10.26508/lsa.202402949)

Corresponding author(s): Wei Li, Capital Medical University

Review Timeline:

Submission Date:	2024-07-17
Editorial Decision:	2024-09-26
Revision Received:	2024-11-06
Editorial Decision:	2024-11-27
Revision Received:	2024-11-30
Accepted:	2024-12-02

Transaction Report:

September 26, 2024

Re: Life Science Alliance manuscript #LSA-2024-02949-T

Xuanshi Liu
Beijing Children's Hospital, Capital Medical University

Dear Dr. Liu,

Thank you for submitting your manuscript entitled "Systematic Assessment of Structural Variant Annotation Tools for Genomic Interpretation" to Life Science Alliance. The manuscript was assessed by expert reviewers, whose comments are appended to this letter. We invite you to submit a revised manuscript addressing the Reviewer comments.

Thank you for this interesting contribution to Life Science Alliance. We are looking forward to receiving your revised manuscript.

Sincerely,

B. MANUSCRIPT ORGANIZATION AND FORMATTING:

Reviewer #1 (Comments to the Authors (Required)):

This study provides a comprehensive benchmarking of eight commonly used structural variant (SV) annotation tools. SVs can have a significant impact on human phenotypes and disease, and are therefore of importance for genomic medicine approaches - but analysing them is complex. The eight tools analysed were divided into two groups: knowledge-driven (AnnotSV, ClassifyCNV) and data-driven (CADD-SV, dbCNV, StrVCTVRE, SVScore, TADA, XCNV). This study comprehensively evaluated the accuracy, robustness and usability of these tools in different genomic contexts and for different biological mechanisms using several curated datasets. The results showed comparable performance between the two groups, but emphasised that the selection of tools should be specific to the respective research purpose. This is a nicely written paper, and I do emphasise the importance of conducting comprehensive benchmarking of SV annotation tools, under various contexts, to provide guidance for biomedical research as well as clinical applications of genomic medicine - and also as a basis for future enhancement of SV annotation tools. I therefore general do find this manuscript and the resource it provides valuable for the genetics community. Comments provided below are meant to further improve this paper.

- The authors have selected eight tools, out of ~20 published to date for SV annotation. What I was missing was a clearer reasoning why these 8 were chosen? The authors state that availability, periodic updates and capacity were the main reasons. Could the authors elaborate on that more? Were only the 8 tools in questions regularly updated since their publication? A new Supplementary mentioning all tools in scope, and clarifying why the 8 here used were chosen, would be a helpful addition to this work.

- I encourage the authors to further expand on their discussion of how their insights could be used to further improve SV annotation in the future. Could future tools benefit from insights from knowledge- and data-driven techniques in an appropriate manner? Could the advent of routine long-read sequencing providing access to regions of the genome previously largely "left out" lead to future challenges?

- The positive and negative sets of SVs for SV annotation tool annotation appear to be valuable. I though encourage the authors to be more detailed with respect to procedures used to generate these sets of data, which future studies may wish to expand from. For example, were there challenges in lifting over SV calls between reference genome assemblies? A Supplementary Table listing all the data sources and references used to generate the data in table 2 is needed, to allow for maximum reproducibility.

- In Figure 3, several axis were unlabelled making it difficult to appreciate what is shown in several panels - including pane C and E. What information is shown on the X-axis of panel B?

- I am not sure about the value of Fig. 5. The authors could consider replacing it with Table S5 which appears to have more utility

Minor comments:

- Introduction: Lappalainen et al. 2019 is not really an SV paper. I suggest adding a more meaningful citation, or replace the citation, with an original publication on SVs.
- Please provide the respective references for ClinVar, Decipher, DGV, GnomAD and 1KG where these resources/tools are first mentioned in this manuscript.
- P. 15: I do not understand what was meant with the following sentence: "This may be due to the fact that the training sets and feature selection for model construction."
- P.16: Please rephrase the following fuzzy sentence: "In noncoding SVs, we observed significant influence on gene regulation despite the absence of protein sequence alterations"
- Do any of the tools tested support the CHM13/T2T genome assembly? Would working on this assembly represent a particular challenge?

Reviewer #2 (Comments to the Authors (Required)):

In this paper by Liu et al, authors explore the performance of various computational tools in predicting the pathogenicity of structural variants. Accurate prediction of the pathogenicity of SVs is an extremely important problem and will have major implications for studying these variants in diseases. As noted by authors there are multiple methods developed for such a task using different approaches. While independent evaluation of these tools is important and commendable, however, there are several drawbacks that reduce enthusiasm for this work.

Major comments:

First, the evaluation dataset needs to be very carefully considered such that none of the SVs used in the evaluation were not also being used in the training of any of these models.

Such an overlap would give an artificial boost to methods that have used them. For example, the StrVCTVRE uses the Gnomad set of SVs in the training set. The authors need to be sure none of the SVs they considered in their evaluation was among these SVs used in the training.

Second, some of the tools use biological annotations (e.g., gene annotation) in their score while others might not. This is another caveat that might artificially give a boost to some of the methods.

Third, there is a large set of known neurodevelopmental associated SVs (mostly denovo) that authors also can use in their evaluation.

Finally, it would be nice to see how the predicted non-coding pathogenic SVs predicted by most tools are related to the 3D genome interactions such as Hi-C data.

Minor comment:

In Figure 4, the for non-coding panel the result for STrVctVre seems missing.

We thank you and the reviewers for evaluating our manuscript. In the revised version, we have included a summary blurb, “This study benchmarks eight structural variant prioritization tools, highlighting their comparable effectiveness in predicting pathogenicity and providing insights for improved genomic research”, and we have reorganized and formatted the manuscript according to the journal’s guidelines.

We have addressed all the points raised by the reviewers and made revisions accordingly. Following the reviewers’ suggestions, we conducted additional analyses and added 2 supplementary tables (Table S1 and S2) in the revised manuscript. The original Figure 5 has been replaced by Table 3 as Reviewer 1 suggested. The new inputs are marked in the pages and lines of the revised clean manuscript. On the following pages, you will find our detailed responses to their comments. We appreciate the positive feedback highlighting the merits of our study and hope that you and the reviewers will be satisfied with the revisions.

Response to reviewer 1

Reviewer #1 (Comments to the Authors (Required))

This study provides a comprehensive benchmarking of eight commonly used structural variant (SV) annotation tools. SVs can have a significant impact on human phenotypes and disease, and are therefore of importance for genomic medicine approaches - but analysing them is complex. The eight tools analysed were divided into two groups: knowledge-driven (AnnotSV, ClassifyCNV) and data-driven (CADD-SV, dbCNV, StrVCTVRE, SVScore, TADA, XCNV). This study comprehensively evaluated the accuracy, robustness and usability of these tools in different genomic contexts and for different biological mechanisms using several curated datasets. The results showed comparable performance between the two groups, but emphasised that the selection of tools should be specific to the respective research purpose. This is a nicely written paper, and I do emphasise the importance of conducting comprehensive benchmarking of SV annotation tools, under various contexts, to provide guidance for biomedical research as well as clinical applications of genomic medicine - and also as a basis for future enhancement of SV annotation tools. I therefore general do find this manuscript and the resource it provides valuable for the genetics community. Comments provided below are meant to further improve this paper.

1. The authors have selected eight tools, out of ~20 published to date for SV annotation. What I was missing was a clearer reasoning why these 8 were chosen? The authors state that availability, periodic updates and capacity were the main reasons. Could the authors elaborate on that more? Were only the 8 tools in questions regularly updated since their publication? A new Supplementary mentioning all tools in scope, and clarifying why the 8 here used were chosen, would be a helpful addition to this work.

Response: At the time we prepared this benchmarking project, we selected 8 tools from 27 published SV prioritization tools based on the following criteria: 1) Availability and periodic updates. We prioritized tools that are publicly available and with enough support in installation, ensuring that they are reliable for current studies. 2) Capacity to handle various SV types without additional information or manual work. The selected tools are capable of handling at least deletions and duplications (mostly for germline variations) which are the most frequent types of SVs without clinical phenotypes or bam files. 3) Computational efficiency and ease of use. Given the large number of SV data in studies, we also considered tools that are efficient in terms of computational resource usage and compatibility with standard pipelines. In revision, we added the selection criteria in the text (P.4 L.72-76) and a summary of all 27 tools evaluated in Supplementary Table S1, highlighting the rationale for selecting these 8 tools.

2. I encourage the authors to further expand on their discussion of how their insights could be used to further improve SV annotation in the future. Could future tools benefit from insights from knowledge- and data-driven techniques in an appropriate manner? Could the advent of routine long-read sequencing providing access to regions of the genome previously largely "left out" lead to future challenges?

Response: We appreciate the reviewer's suggestion to expand on the discussion. We believe that the insights from our study highlight the importance of merging both knowledge-based and data-driven approaches to improve future SV annotation tools. Additionally, the advancement of long-read sequencing will have both new opportunities and challenges for SV detection and annotation. We have expanded a more detailed discussion as the following in the revised manuscript (P.13 L.276-283 and P.15 L.305-314).

Knowledge- and data-driven techniques. Additionally, our benchmarking study highlights the strengths and limitations of both knowledge-driven and data-driven techniques. Future tools could benefit from a hybrid approach. Knowledge-based techniques, which leverage existing knowledge and framework like the ACMG guidelines, are essential for determine the pathogenicity of SVs. Incorporating data-driven techniques can be highly beneficial in identifying novel or potentially pathogenic SVs that may not be well understood yet. Integrating both approaches can lead to more comprehensive and accurate SV prioritization, especially for novel or complex regions.

Challenges and opportunities with long-read sequencing. The increasing accessibility of long-read sequencing opens up new opportunities for SV detection. This technique is particularly effective for identifying complex SVs, repetitive regions, and resolving large structural changes that short-read technologies failed. However, it also faces challenges. These new regions will require updated annotations and retraining of data-driven models to handle the unique properties of long-read data. Moreover, integrating long-read sequencing data with the existing short-read data and annotations poses another challenge. There is a need for tools that can efficiently combine information from multiple sequencing platforms and provide a unified annotation framework.

3 The positive and negative sets of SVs for SV annotation tool annotation appear to be valuable. I though encourage the authors to be more detailed with respect to procedures used to generate these sets of data, which future studies may wish to expand from. For example, were there challenges in lifting over SV calls between reference genome assemblies? A Supplementary Table listing all the data sources and references used to generate the data in table 2 is needed, to allow for maximum reproducibility.

Response: We appreciate the reviewer's suggestion. We have provided a

comprehensive breakdown of the sources and references for each dataset in Supplementary Table S2. This includes details about the datasets used in Table 2 with additional references, data sources, and procedures. This ensures full reproducibility of the results and facilitates future expansion by other studies.

Regarding challenges, we did face issues while converting SVs between reference genome assemblies, particularly when lifting SVs from hg38 to hg19. We used UCSC liftover to convert SVs for datasets like GWAS SVs, eQTL SVs, and COSMIC somatic mutations. However, a small number of SVs (ranging from 1 to 2 per dataset) failed to map to hg19 due to incompatibilities, and we excluded these from further analysis.

4 In Figure 3, several axis were unlabelled making it difficult to appreciate what is shown in several panels - including pane C and E. What information is shown on the X-axis of panel B?

Response: We appreciate the reviewer's comments, and we have updated the labelling of axis in Fig. 3B, C, E accordingly.

5 I am not sure about the value of Fig. 5. The authors could consider replacing it with Table S5 which appears to have more utility

Response: Thanks for the valuable comments. We have replaced Figure 5 with Table 3 in the revised manuscript.

Minor comments:

1 Introduction: Lappalainen et al. 2019 is not really an SV paper. I suggest adding a more meaningful citation, or replace the citation, with an original publication on SVs.

Response: We have included studies from gnomAD and Icelander cohorts, which represented short read and long read sequencing studies. We have updated the description accordingly (P.3 L.57-60). "Additionally, the vast number of SVs detected, thousands through short-read and up to 20,000 through long-read whole genome sequencing (WGS) (Collins et al. 2020; Beyter et al. 2021), results in the complexity of their analysis and interpretation."

2 Please provide the respective references for ClinVar, Decipher, DGV, GnomAD and 1KG where these resources/tools are first mentioned in this manuscript.

Response: We have included the references in the revised manuscript (P.5 L.91-95). "In contrast, data-driven approaches based their training sets and features on gold standard datasets, including ClinVar (Landrum et al. 2016),

Decipher (Firth et al. 2009), DGV (MacDonald et al. 2014), GnomAD (Collins et al. 2020), and 1KG (Auton et al. 2015) , with a focus on specific aspects of SV analysis”

3 P. 15: I do not understand what was meant with the following sentence: "This may be due to the fact that the training sets and feature selection for model construction."

Response: We have revised the description accordingly (P.9 L.175-177). “This may be due to the smaller number of duplications in the training set and feature selection processes that were more tailored to deletions.”

4 P.16: Please rephrase the following fuzzy sentence: "In noncoding SVs, we observed significant influence on gene regulation despite the absence of protein sequence alterations"

Response: We have rephrased the sentence accordingly (P.10. L.207-208). “In noncoding SVs, we observed that TADA, SVScore, and AnnotSV were the top performers, demonstrating high AUC values of 0.92, 0.86, and 0.83, respectively”.

5 Do any of the tools tested support the CHM13/T2T genome assembly? Would working on this assembly represent a particular challenge?

Response: We appreciate the reviewer's question regarding the CHM13/T2T genome assembly. The CHM13/T2T represents a major improvement in genome completeness, especially in difficult regions like centromeres and telomeres, making it a valuable resource for future SV studies.

Most of the tools we tested were designed for assemblies like hg19 or hg38 and no tools support CHM13/T2T specifically. A key challenge in using CHM13/T2T is that tool development requires annotation and resources such as population frequency and gene annotations. All these resources are better developed at hg19 or hg38. Therefore, adapting to CHM13/T2T would require updating all these resources.

We have included this point in the discussion of our revised manuscript, acknowledging the potential challenges and benefits of working with CHM13/T2T (P.15 L.325-328). “Finally, CHM13/T2T represents a major improvement in genome completeness, especially in difficult regions like centromeres and telomeres. Combining it with updated annotations and resources could be a promising direction for tool development, benefiting future clinical and biological studies.”

Response to reviewer 2

Reviewer #2 (Comments to the Authors (Required))

In this paper by Liu et al, authors explore the performance of various computational tools in predicting the pathogenicity of structural variants. Accurate prediction of the pathogenicity of SVs is an extremely important problem and will have major implications for studying these variants in diseases. As noted by authors there are multiple methods developed for such a task using different approaches. While independent evaluation of these tools is important and commendable, however, there are several drawbacks that reduce enthusiasm for this work.

Major comments:

1. First, the evaluation dataset needs to be very carefully considered such that none of the SVs used in the evaluation were not also being used in the training of any of these models.

Such an overlap would give an artificial boost to methods that have used them. For example, the StrVCTVRE uses the Gnomad set of SVs in the training set. The authors need to be sure none of the SVs they considered in their evaluation was among these SVs used in the training.

Response: We fully agree that ensuring no overlap between training and test sets is critical to avoid artificially inflating performance metrics. This was a key consideration in the design of our study. As mentioned in the manuscript, we employed two approaches to avoid any overlap between the training sets used by the tools and the test sets in our benchmark analysis. Firstly, wherever possible, we selected datasets with publication dates later than the date when the respective tools were finalized and submitted for publication. This helps minimize the risk of including data that might have been used in training. Secondly, we used "bedtools" to systematically eliminate any overlapping SVs between the training and test sets, as well as between positive and negative test sets. For instance, in the case of StrVCTVRE, which utilized gnomAD V2 for training, we excluded all overlapping SVs between gnomAD V4.1 (our test set) and gnomAD V2 by employing the "bedtools intersect -v" command.

2. Second, some of the tools use biological annotations (e.g., gene annotation) in their score while others might not. This is another caveat that might artificially give a boost to some of the methods.

Response: We sincerely appreciate the reviewer's thoughtful comments and concerns. We understand the point raised regarding the potential discrepancies caused by some tools incorporating biological annotations (e.g.,

gene annotations) while others do not. However, we would like to clarify the rationale behind our study and the intent of our comparisons.

In the field of SV pathogenicity prediction, there are various types of methods available including knowledge-based, and data-driven or model-based. These methods rely on different assumptions and approaches, which reflects the diversity in this evolving area. This diversity also poses the challenge to determine which tools perform better under certain conditions. It is exactly this uncertainty that motivated us to conduct this comparison.

Our primary goal is not to identify a single "best" tool but rather to explore how different methods perform in scoring SVs under the current state of knowledge. By doing so, we aim to provide computational scientists with insights into the strengths and weaknesses of various approaches and offer biomedical scientists practical guidance for selecting appropriate tools based on their specific research needs and scenarios.

We believe that this broader, comparative approach is necessary and valuable for fostering further advancements in both computational tool development and clinical research. Nevertheless, we acknowledge the limitations associated with the use of different sources of information, and we aimed to control for such discrepancies as much as possible in our evaluations by using the identical datasets covering different biological insights.

3. Third, there is a large set of known neurodevelopmental associated SVs (mostly de novo) that authors also can use in their evaluation.

Response: We found 94 CNVs from Gene4Denovo (Zhao et al. 2019), which compiles de novo mutations linked to diseases or discovered in patients from published studies. These CNVs, with a mean length of 110 bp and a standard deviation of 54.8, were categorized into a positive set. To create a negative set, we randomly selected type- and length-matched rare CNVs from the gnomAD V4.1 database.

We applied the same pipeline as in our original analysis and found that SVScore ranked first in performance, followed by CADD-SV (Rebuttal Figure 1). SVScore's top performance in this context indicates that it could be particularly useful for identifying pathogenic variants where data is sparse. However, our comparison of performance across different lengths of CNVs reveals significant differences. Since these de novo CNVs are much shorter than most of CNVs (average length is 287,404 bp) in other datasets, we have chosen not to include them in the main text.

Rebuttal Figure 1. Performance over six different datasets covering various biological mechanisms including noncoding SVs, long range SVs, somatic SVs, GWAS SV, eQTL SV and de novo SV. AUC: area under the curve; SV: structural variant.

4. Finally, it would be nice to see how the predicted non-coding pathogenic SVs predicted by most tools are related to the 3D genome interactions such as Hi-C data.

Response: We appreciate the reviewer’s insightful suggestion to integrate 3D genome interactions, such as Hi-C data, into the analysis of predicted non-coding pathogenic SVs. Incorporating 3D genomic context can indeed provide valuable insights, and we have explored two key approaches in this regard.

Firstly, disruptions in topologically associating domain (TAD) boundaries are known to impact gene regulation. Among the tools we evaluated, TADA specifically addresses this aspect by combining TAD boundary information with functional annotations to assess the pathogenicity of SVs through the lens of 3D genome interactions. This approach allowed us to explore the effects of TAD disruptions on SV pathogenicity in more detail. For example, boundary stability is an important consideration. If a SVs occurs near a stable TAD

boundary, it may have a greater impact on the regulation of gene expression, as variations in these regions are more likely to lead to changes in gene expression patterns (Rebuttal Table 1).

Rebuttal Table 1. Evaluation of topologically associating domain (TAD) boundaries.

Noncoding SVs	Number of affected Genes	Number of affected Enhancers	Boundary Distance	Boundary Stability
chr5:14015350-14055499	0	7	55350	0.986859
chr11:131345836-131583798	2	6	0	0.855772
chr14:43545282-43837068	0	2	985282	0.715939
chr15:25257218-25375375	0	0	64625	0.168811
chr16:6908075-7079700	1	3	188075	0.887679
chr17:68680882-68965323	0	3	40882	0.946567

Secondly, promoter enhancer interactions provide another critical layer of 3D genomic organization. To investigate this, we utilized TADeus2 (Poszewiecka et al. 2022), a web based tool designed to evaluate potential promoter-enhancer links and predict target genes. Below is an example of our findings, showing noncoding SVs and their related gene targets, along with enhancer promoter interaction data and pathogenicity scores (Rebuttal Table 2).

Rebuttal Table 2. Evaluation of promoter-enhancer interactions.

Noncoding SVs	Gene symbol	Enhancer-promoter interactions number	Distance from breakpoints	Total pathogenicity score
chr5:14015350-14055499	DNAH5	24	43572	4
	TRIO	20	128101	4
chr11:131345836-131583798	NTM	17	105,464	2
	OPCML	0	939,035	2
chr14:43545282-43837068	HNRNPUP1	0	67,270	1
	TUBBP3	0	92,168	1
chr15:25257218-25375375	SNRPN	6	151,505	3
	UBE3A	0	308,823	3
chr16:6908075-7079700	RBFOX1	1	683,641	2
	ABAT	0	1,754,876	2
chr17:68680882-68965323	KCNJ2	0	516,068	2
	KCNJ16	0	631,312	2

We have included this point in the discussion, highlighting the two key approaches (P.15. L.319-324). “A particular challenge lies in interpreting the biological significance of SVs within non-coding regions, where their impact often depends on disruptions to regulatory elements such as enhancer-promoter interactions and topologically associating domain (TAD) boundaries. Tools that incorporate 3D genomic context could improve non-coding SV interpretation (Hertzberg et al. 2022; Poszewiecka et al. 2022).”

Minor comment:

In Figure 4, the for non-coding panel the result for STrVctVre seems missing.

Response: We thank the reviewer for pointing out this lack of clarity. There is no non-coding panel for StrVCTVRE giving that the model behind StrVCTVRE was built on coding regions, as we mentioned it at the manuscript, “However, it's important to note that CADD-SV and StrVCTVRE were not applicable for noncoding SVs due to their focus on protein-coding genes.”

References:

- Auton A, Brooks LD, Durbin RM, Garrison EP, Kang HM, Korbel JO, Marchini JL, McCarthy S, McVean GA, Abecasis GR. 2015. A global reference for human genetic variation. *Nature* **526**: 68-74.
- Collins RL, Brand H, Karczewski KJ, Zhao X, Alföldi J, Francioli LC, Khera AV, Lowther C, Gauthier LD, Wang H et al. 2020. A structural variation reference for medical and population genetics. *Nature* **581**: 444-451.
- Firth HV, Richards SM, Bevan AP, Clayton S, Corpas M, Rajan D, Vooren SV, Moreau Y, Pettett RM, Carter NP. 2009. DECIPHER: Database of Chromosomal Imbalance and Phenotype in Humans Using Ensembl Resources. *The American Journal of Human Genetics* **84**: 524-533.
- Hertzberg J, Mundlos S, Vingron M, Gallone G. 2022. TADA-a machine learning tool for functional annotation-based prioritisation of pathogenic CNVs. *Genome Biol* **23**: 67.
- Landrum MJ, Lee JM, Benson M, Brown G, Chao C, Chitipiralla S, Gu B, Hart J, Hoffman D,

Hoover J et al. 2016. ClinVar: public archive of interpretations of clinically relevant variants. *Nucleic Acids Res* **44**: D862-868.

MacDonald JR, Ziman R, Yuen RK, Feuk L, Scherer SW. 2014. The Database of Genomic Variants: a curated collection of structural variation in the human genome. *Nucleic Acids Res* **42**: D986-992.

Poszewiecka B, Pienkowski VM, Nowosad K, Robin JD, Gogolewski K, Gambin A. 2022. TADeus2: a web server facilitating the clinical diagnosis by pathogenicity assessment of structural variations disarranging 3D chromatin structure. *Nucleic Acids Res* **50**: W744-W752.

Zhao G, Li K, Li B, Wang Z, Fang Z, Wang X, Zhang Y, Luo T, Zhou Q, Wang L et al. 2019. Gene4Denovo: an integrated database and analytic platform for de novo mutations in humans. *Nucleic Acids Research* doi:10.1093/nar/gkz923.

November 27, 2024

RE: Life Science Alliance Manuscript #LSA-2024-02949-TR

Prof. Wei Li
Capital Medical University
Beijing Children's Hospital
56 Nan-Li-Shi Road
Xicheng District, Beijing 100045
China

Dear Dr. Li,

Thank you for submitting your revised manuscript entitled "Systematic Assessment of Structural Variant Annotation Tools for Genomic Interpretation". We would be happy to publish your paper in Life Science Alliance pending final revisions necessary to meet our formatting guidelines.

- please be sure that the authorship listing and order is correct
- please add the Twitter handle of your host institute/organization as well as your own or/and one of the authors in our system

LSA now encourages authors to provide a 30-60 second video where the study is briefly explained. We will use these videos on social media to promote the published paper and the presenting author (for examples, see <https://docs.google.com/document/d/1-UWCfbE4pGcDdcgzcmiuJI2XMBJnxKYeqRvLLrLS08s/edit?usp=sharing>). Corresponding or first-authors are welcome to submit the video. Please submit only one video per manuscript. The video can be emailed to contact@life-science-alliance.org

A. FINAL FILES:

B. MANUSCRIPT ORGANIZATION AND FORMATTING:

Thank you for your attention to these final processing requirements. Please revise and format the manuscript and upload materials within 5 days.

Sincerely,

Reviewer #2 (Comments to the Authors (Required)):

I have no further comments. All my concerns have been addressed in the revised manuscript.

December 2, 2024

RE: Life Science Alliance Manuscript #LSA-2024-02949-TRR

Prof. Wei Li
Capital Medical University
Beijing Children's Hospital
56 Nan-Li-Shi Road
Xicheng District, Beijing 100045
China

Dear Dr. Li,

Thank you for submitting your Resource entitled "Systematic Assessment of Structural Variant Annotation Tools for Genomic Interpretation". It is a pleasure to let you know that your manuscript is now accepted for publication in Life Science Alliance. Congratulations on this interesting work.

DISTRIBUTION OF MATERIALS:

Again, congratulations on a very nice paper. I hope you found the review process to be constructive and are pleased with how the manuscript was handled editorially. We look forward to future exciting submissions from your lab.

Sincerely,
